# HallE-Switch: Rethinking and Controlling Object Existence Hallucinations in Large Vision-Language Models for Detailed Caption

## Abstract

Current large vision-language models (LVLMs) achieve remarkable progress, yet there remains significant uncertainty regarding their ability to accurately apprehend visual details, that is, in performing detailed captioning. To address this, we introduce *CCEval*, a GPT-4 assisted evaluation method tailored for detailed captioning. Interestingly, while LVLMs demonstrate minimal object existence hallucination in existing VQA benchmarks, our proposed evaluation reveals continued susceptibility to such hallucinations. In this paper, we make the first attempt to investigate such hallucination from different aspects, including image resolution, the language decoder size, and instruction data amount, quality, granularity. Our findings underscore the unwarranted inference when the language description includes details at a finer object granularity than what the vision module can ground or verify, thus inducing hallucination. To control such hallucinations, we further attribute the reliability of captioning to contextual knowledge (involving only contextually grounded objects) and parametric knowledge (containing inferred objects by the model). Thus, we introduce *HallE-Switch*, a controllable LVLM in terms of **Hall**ucination in object **E**xistence. HallE-Switch can condition the captioning to shift between (i) exclusively depicting contextual knowledge for grounded objects and (ii) blending it with parametric knowledge to imagine inferred objects. Our method reduces hallucination by 44% compared to LLaVA$_{7B}$ and maintains the same object coverage.

## 1 Introduction

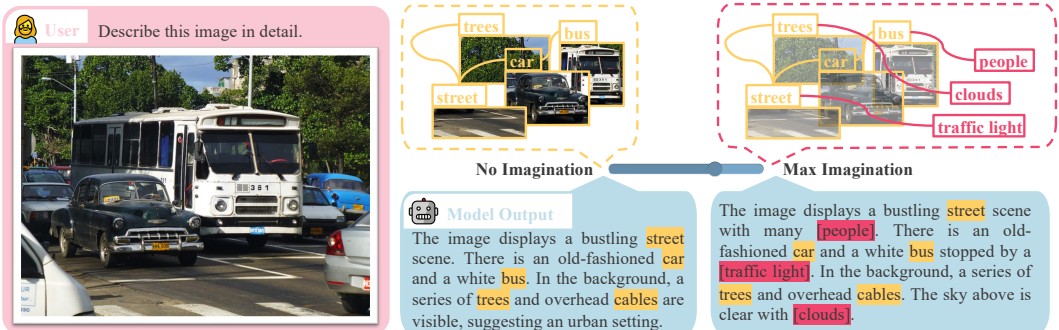

Figure 1: The figure shows that HallE-Switch uses a single continuous parameter during inference to control imagination in the outputted caption. A switch value of "−1" makes the model use solely contextual knowledge (visually grounded objects), such as trees, buses, cars, and streets. A switch value of "+1" makes the model incorporate parametric knowledge (inferred objects), such as people, clouds, and traffic lights, with the [object] marker labeling those inferred objects.

In recent years, Large Vision-Language Models (LVLMs) (Liu et al., 2023d; Dai et al., 2023; Li et al., 2023b; Zhu et al., 2023) have achieved significant progress, advancing tasks such as detailed

captioning, visual conversations, and vision question-answering (VQA) (Goyal et al., 2017; Liu et al., 2023e; Hudson & Manning, 2019; Fu et al., 2023). However, similar to Large Language Models (LLMs) (Touvron et al., 2023a; Team, 2023; OpenAI, 2022) in the NLP domain, LVLMs confront the issue of hallucination. This is particularly severe in detailed image captioning, which hinders the performance of downstream applications in robotics (Huang et al., 2023), visual search (Hu et al., 2023), etc. To better understand and address this challenge, we first outline three types of object hallucinations frequently observed in the detailed captions: (1) Object Existence Hallucination - The detailed image description references objects that are not present; (2) Object Attribute Hallucination - The detailed image description inaccurately characterizes objects, misrepresenting attributes such as color, shape, and size; (3) Object Relationship Hallucination - The detailed image description inaccurately depicts the relationships or interactions among objects, including erroneous relative positions, interaction states, and actions involving two or more objects. In this work, we mainly focus on defining the metric, analyzing the cause, and addressing the the problem of **object existence hallucination**.

Evaluating detailed captions is inherently complex. Some of the efforts, including benchmarks like POPE (Li et al., 2023e), evaluate object hallucination using VQA. Such a bias towards VQA-based evaluations might result in an incomplete assessment of detailed captions, which requires obtaining a comprehensive view of visual details. To bridge this gap, we introduce *CCEval*, designed specifically for object existence hallucination in detailed captions. To avoid the model gaining an unfair advantage by favoring shorter descriptions, CCEval maintains consistency in metrics such as average sentence length and the number of objects. Notably, even models that well-performed on VQA-based object hallucination benchmarks showed substantial hallucinations when evaluated with CCEval.

In our exploration to uncover the underlying cause of object existence hallucination, we look into various factors including the size of the language decoder, the quantity, quality, and granularity of instruction data, and the input resolution to the vision encoder. We conclude the most crucial factor to be the alignment between objects mentioned in training caption and those vision encoder can perceive. During training of LVLMs, the goal is to establish a one-to-one correspondence between objects mentioned in the caption and those present in the image. Objects successfully grounded by the vision encoder form accurate associations, internalizing them as **contextual knowledge**. Conversely, objects in the language that the vision encoder fails to ground create word-word semantic associations, which can be attributed to the generalization from the **parametric knowledge** within the model's parameters. During inference, when the model draws from such parametric knowledge, any misalignment can manifest as hallucination, as the model attempts to "guess" details not grounded by the vision module.

To address such hallucination, we are motivated by that *not all hallucination is bad, and it is more desirable to control the generalization rather than outright removal of all imagined objects.* Recognizing the significance of both contextual and parametric knowledge in ensuring generation reliability, we present *HallE-Switch*, a novel approach to control the extent of expressed hallucination or parametric knowledge. We curate a 33k dataset similar to LLaVA (Liu et al., 2023d), incorporating both pure contextual knowledge and a blend of contextual knowledge with marked parametric knowledge. Leveraging this dataset, we train a lightweighted single linear layer to control over the frozen LVLM. As demonstrated in Figure 1, a singular continuous parameter adjustment (e.g. $-1 \rightarrow +1$) during inference enables the model to produce detailed captions with only contextual knowledge (e.g., $-1$) or blend with parametric knowledge (e.g., $+1$). Furthermore, the inferred objects from parametric knowledge are automatically highlighted with distinct tokens (e.g., `[object]`) for human reference. This method offers the advantage of preserving object count and coverage as well as sentence length, while effectively control object existence hallucination.

Overall, our contributions are:

- A novel evaluation method for detailed caption object existence hallucination, with metrics such as object count, coverage, and average sentence length, alongside hallucination assessment;

- A comprehensive analysis on LVLM components that influence hallucination, with a specific focus on alignment issues in the vision encoder and instruction data;

- A first approach to control object existence hallucination within detailed captions.

Table 1: We evaluate LLaVA Vicuna$_{7B}$, LLaVA Vicuna$_{13B}$, Shikra$_{7B}$, InstructBLIP Vicuna$_{7B}$ public checkpoints on VQA-based benchmarks, including POPE and MME.

| Benchmark | Model | Accuracy↑ | Precision↑ | Recall↑ | F1↑ | Yes (%) |
|---|---|---|---|---|---|---|
| POPE - Random | LLaVA$_{7B}$ | 73.13 | 66.95 | 94.53 | 78.39 | 72.78 |
| | LLaVA$_{13B}$ | 78.49 | 73.57 | 90.93 | 81.34 | 63.71 |
| | Shikra$_{7B}$ | 86.99 | 94.77 | 79.13 | 86.25 | 43.04 |
| | InstructBLIP$_{7B}$ | 86.60 | 80.74 | 96.13 | 87.77 | 59.53 |
| POPE - Popular | LLaVA$_{7B}$ | 59.87 | 55.88 | 93.80 | 70.03 | 83.93 |
| | LLaVA$_{13B}$ | 70.80 | 64.73 | 91.40 | 75.79 | 70.60 |
| | Shikra$_{7B}$ | 84.35 | 88.10 | 79.43 | 83.54 | 45.08 |
| | InstructBLIP$_{7B}$ | 71.27 | 64.20 | 96.13 | 76.99 | 74.87 |
| POPE - Adversarial | LLaVA$_{7B}$ | 57.06 | 54.07 | 93.93 | 68.63 | 86.87 |
| | LLaVA$_{13B}$ | 63.93 | 59.03 | 91.07 | 71.63 | 77.13 |
| | Shikra$_{7B}$ | 82.88 | 85.32 | 79.43 | 82.27 | 46.55 |
| | InstructBLIP$_{7B}$ | 72.10 | 65.13 | 95.13 | 77.32 | 73.03 |
| Benchmark | Model | Existence↑ | Count↑ | Position↑ | Color↑ | Total↑ |
| MME | LLaVA$_{7B}$ | 150.00 | 48.33 | 50.00 | 55.00 | 303.33 |
| | LLaVA$_{13B}$ | 180.00 | 113.33 | 55.00 | 95.00 | 443.33 |
| | Shikra$_{7B}$ | 185.00 | 118.33 | 75.00 | 155.00 | 533.33 |
| | InstructBLIP$_{7B}$ | 185.00 | 60.00 | 50.00 | 125.00 | 420.00 |

Table 2: Comparison between CHAIR and our evaluation method, CCEval.

| Model | CHAIR | | | | CCEval (Ours) | | | | |
|---|---|---|---|---|---|---|---|---|---|
| | $CHAIR_i$↓ | $CHAIR_s$↓ | Avg. Length↑ | Avg. Object↑ | $CHAIR_i$↓ | $CHAIR_s$↓ | Coverage↑ | Avg. Length↑ | Avg. Object↑ |
| LLaVA$_{7B}$ | 24.1 | 9.1 | 42.5 | 3.7 | 72.00 | 19.7 | 32.74 | 92.27 | 9.19 |
| LLaVA$_{13B}$ | 60.6 | 18.4 | 90.2 | 7.6 | 79.00 | 23.80 | 33.56 | 108.02 | 9.28 |
| Shikra$_{7B}$ | 59.1 | 16.6 | 91.2 | 7.5 | 83.00 | 24.40 | 33.29 | 109.37 | 9.10 |
| InstructBLIP$_{7B}$ | 1.4 | 1.7 | 2.3 | 0.8 | 72.00 | 22.30 | 29.76 | 108.42 | 8.04 |

## 2 HALLUCINATION ANALYSIS

Object existence hallucination can be influenced by several factors, including the language decoder, instruction data, and vision encoder. In our analysis, we address each factor individually. For a diverse methodological exploration, we select LLaVA, InstructBLIP (Dai et al., 2023), and Shikra (Chen et al., 2023): LLaVA and Shikra share the same model structure; Shikra and InstructBLIP use mixed-dataset and multi-task instruction data; InstructBLIP finetunes only Q-former, while the other finetune projector and LLM. More details about models can be found in Appendix.

### 2.1 BENCHMARKS

There are two primary approach, VQA-based and caption-based benchmarks, for evaluating object existence hallucination in LVLMs.

**VQA-based benchmarks** pose questions about objects within images. For a model to be considered hallucination-free, it should address these visual questions accurately. Notably, a large proportion of questions are simply binary, typically asking about the presence or attributes of objects.

The POPE benchmark evaluates object existence hallucination by a polling-based query method, consisting of a series of yes/no questions on sampled objects from visual instructions. POPE contains three sets: random, popular, and adversarial. These subsets respectively focus on randomly selected objects, frequently occurring objects, and those objects that co-occur in training sets. We choose POPE evaluation on MSCOCO (Lin et al., 2014) dataset. MME (Fu et al., 2023) coarse-grained recognition construct yes/no questions similarly but selects objects at random. This benchmark has 30 images, with each image paired with two questions: one positive and one negative.

In Table 1, LLaVA$_{7B}$ exhibits the greatest degree of hallucination, whereas Shikra outperforms other models in both POPE and MME. Specifically, Shikra shows a significantly higher F1 score in both POPE-popular and POPE-adversarial categories, while LLaVA$_{7B}$ displays the lowest. Additionally, Shikra's "Yes" ratio is closer to a balanced 50% compared to other models. However, in subsequent sections, we demonstrate that these observations from VQA-based benchmarks are not consistent with those from caption-based benchmarks.

Table 3: Performance of LLaVA and InstructBLIP with different sizes of language decoder. LLaVA are trained on CC-595k for stage one and Instruction-150k for stage two.

| Benchmark | Model | Accuracy↑ | Precision↑ | Recall↑ | F1↑ | Yes (%) |
|---|---|---|---|---|---|---|
| POPE - Random | LLaVA$_{7B}$ | 75.77 | 69.79 | 93.47 | 79.91 | 69.04 |
| | LLaVA$_{13B}$ | 78.49 | 73.57 | 90.93 | 81.34 | 63.71 |
| | LLaVA$_{33B}$ | 78.14 | 73.18 | 90.93 | 81.09 | 64.05 |
| | InstructBLIP$_{7B}$ | 86.60 | 80.74 | 96.13 | 87.77 | 59.53 |
| | InstructBLIP$_{13B}$ | 88.73 | 86.67 | 92.33 | 89.41 | 54.91 |
| POPE - Popular | LLaVA$_{7B}$ | 65.07 | 59.60 | 93.53 | 72.81 | 78.47 |
| | LLaVA$_{13B}$ | 70.80 | 64.73 | 91.40 | 75.79 | 70.60 |
| | LLaVA$_{33B}$ | 72.43 | 66.45 | 90.60 | 76.67 | 68.17 |
| | InstructBLIP$_{7B}$ | 71.27 | 64.20 | 96.13 | 76.99 | 74.87 |
| | InstructBLIP$_{13B}$ | 80.53 | 74.70 | 92.33 | 82.59 | 61.80 |
| POPE - Adversarial | LLaVA$_{7B}$ | 57.07 | 54.07 | 93.93 | 68.63 | 86.87 |
| | LLaVA$_{13B}$ | 63.93 | 59.03 | 91.07 | 71.63 | 77.13 |
| | LLaVA$_{33B}$ | 66.30 | 60.91 | 91.00 | 72.98 | 74.70 |
| | InstructBLIP$_{7B}$ | 72.10 | 65.13 | 95.13 | 77.32 | 73.03 |
| | InstructBLIP$_{13B}$ | 73.97 | 67.53 | 92.33 | 78.01 | 68.37 |

| Benchmark | Model | CHAIR$_s$↓ | CHAIR$_i$↓ | Coverage↑ | Avg. Length↑ | Avg. Object↑ |
|---|---|---|---|---|---|---|
| CCEval (Ours) | LLaVA$_{7B}$ | 82.00 | 25.30 | 33.58 | 109.89 | 9.31 |
| | LLaVA$_{13B}$ | 79.00 | 23.80 | 33.56 | 108.02 | 9.28 |
| | LLaVA$_{33B}$ | 82.00 | 21.80 | 31.26 | 106.85 | 9.07 |
| | InstructBLIP$_{7B}$ | 72.00 | 22.30 | 29.76 | 108.42 | 8.04 |
| | InstructBLIP$_{13B}$ | 64.00 | 16.70 | 33.60 | 101.63 | 8.06 |

**Caption-based benchmarks**, like CHAIR, begin by splitting the sentence and extracting nouns. Subsequently, it augments the ground truth objects by incorporating hard-coded synonyms and phrases, forming a ground truth set. The benchmark then identifies hallucinated objects by comparing the objects in the caption with this ground truth set. CHAIR computes CHAIR$_i$ and CHAIR$_s$ as follows:

$$\text{CHAIR}_i = \frac{|\{\text{hallucinated objects}\}|}{|\{\text{all objects mentioned}\}|}$$

$$\text{CHAIR}_s = \frac{|\{\text{sentences with hallucinated object}\}|}{|\{\text{all sentences}\}|}$$

Table 2(left) reveals that while InstructBLIP exhibits minimal object existence hallucination, it averages a mere 0.8 objects per sentence. In contrast, LLaVA$_{13B}$ and Shikra manifest a higher degree of hallucination, but they also generate more detailed captions, outputting as many as 7.6 and 7.5 objects per sentence, respectively. We find comparing object hallucinations is impractical when there is a significant disparity in average sentence length and the number of objects.

Apart from these disparities, the use of a hard-coded ground truth set is another challenge. To counter these challenges, we introduce *CCEval*, a GPT-4 assisted evaluation for detailed captions. We first prompt LVLMs to generate detailed captions on 100 randomly sampled images from Visual Genome (Krishna et al., 2017). Subsequently, utilizing GPT-4's in-context learning capabilities, we extract individual objects from these captions and identify hallucinated ones by referencing the provided ground truth objects. On top of CHAIR Rohrbach et al. (2018), we introduce "coverage" metric to ensure that the captions are detailed enough. This metric computes the ratio of objects in the caption that match the ground truth to the total number of ground truth objects. We additionally record and balance the average number of objects as well as the average length of captions across all cases. More details on prompts of CCEval can be found in Appendix.

As reflected in Table 2(right), when subjected to consistent constraints—average sentence length approximately 100 words and around 9 objects per sentence—all models exhibit comparably sub-optimal results. Interestingly, while Shikra surpass other models in VQA-based benchmarks, especially in the POPE , it under-performs in CCEval. This suggests that object existence hallucination in detailed captions is not consistently captured by VQA-based evaluations.

## 2.2 Language Decoder

We investigate if expanding the size of the language backbone can mitigate object existence hallucination. As detailed in Table 3, the language decoder of LLaVA is increased from 7B to 33B, and for

Table 4: Performance of LLaVA$_{7B}$ with different sizes of data. 80K and 158K contains 80K and 158K data respectively, and SVIT contains 2.4M.

| Benchmark | Finetune Data | Accuracy↑ | Precision↑ | Recall↑ | F1↑ | Yes (%) |
|---|---|---|---|---|---|---|
| POPE - Random | 80K | 73.13 | 66.95 | 94.53 | 78.39 | 72.78 |
| | 158K | 75.77 | 69.79 | 93.47 | 79.91 | 69.04 |
| | SVIT | 52.34 | 52.00 | 97.87 | 67.92 | 97.01 |
| POPE - Popular | 80K | 59.87 | 55.88 | 93.80 | 70.03 | 83.93 |
| | 158K | 65.07 | 59.60 | 93.53 | 72.81 | 78.47 |
| | SVIT | 50.77 | 50.43 | 90.47 | 64.76 | 89.70 |
| POPE - Adversarial | 80K | 57.07 | 54.07 | 93.93 | 68.63 | 86.87 |
| | 158K | 58.47 | 55.00 | 93.07 | 69.14 | 84.6 |
| | SVIT | 51.37 | 50.77 | 90.33 | 65.00 | 88.97 |

| Benchmark | Finetune Data | $CHAIR_i$↓ | $CHAIR_s$↓ | Coverage↑ | Avg. Length↑ | Avg. Object↑ |
|---|---|---|---|---|---|---|
| CCEval (Ours) | 80K | 72.00 | 19.70 | 32.74 | 92.27 | 9.19 |
| | 158K | 82.00 | 25.30 | 33.58 | 109.89 | 9.31 |
| | SVIT | 87.00 | 23.30 | 47.46 | 296.63 | 18.14 |

Table 5: Performance of LLaVA with Llama 2$_{13B}$ language decoder and CLIP-Large vision encoder with different input resolutions.

| Benchmark | Vision Encoder | $CHAIR_s$↓ | $CHAIR_i$↓ | Coverage↑ | Avg. Length↑ | Avg. Object↑ |
|---|---|---|---|---|---|---|
| CCEval (Ours) | CLIP-L-112x | 79.00 | 21.70 | 32.04 | 110.36 | 9.12 |
| | CLIP-L-224x | 74.00 | 19.30 | 32.83 | 113.03 | 9.18 |
| | CLIP-L-336x | 64.00 | 16.00 | 33.37 | 108.52 | 8.92 |

InstructBLIP, it is increased from 7B to 13B. The result shows that hallucination for LLaVA reduced for POPE but not for CCEval. For InstructBLIP, CHAIR$_i$ and CHAIR$_s$ is reduced by 8 and 5.6 on CCEval, respectively. However, although there is a gain for scaling up language backbone, it is not consistent or salient from the observation, suggesting language decoder is not a primary factor in reducing hallucination.

## 2.3 DATA

Similar to our approach with the language decoder, we begin by scaling up the volume of instruction finetuning data, ranging from 80K to 2.4M. As illustrated in Table 4, the LLaVA$_{7B}$ model, finetuned on 80K instruction data, exhibits fewer object existence hallucinations compared to the models finetuned on 150K and SVIT (Zhao et al., 2023a). The result suggests extra data without quality guarantee may increase hallucination for both VQA-based and caption-based evaluations. Given that all three datasets are generated by GPT-4, we question the quality of the data. LRV also raises this concern, suggesting that the training data itself might contain hallucinations. Some examples of training data are presented in the Appendix. Interestingly, our examination shows no object existence hallucination: the objects in the captions are contained the ground truth objects from MSCOCO. However, we identify certain ground truth objects are challenging for human observers to ground, due to factors like size, resolution, and occlusion. This led us to hypothesize that the vision encoder might also struggle to ground these objects effectively.

## 2.4 VISION ENCODER

Intuitively, increasing image resolution enhances model's perception of finer details, thus making the grounding of objects mentioned in the caption easier. To verify our hypothesis from the previous section, we increase the input image resolution for the vision encoder. Specifically, for our evaluation, the resolution for LLaVA$_{7B}$ was incremented from 224x to full resolution using a sliding

Table 6: Performance of LLaVA$_{7B}$ and with sliding window technique (SW).

| Benchmark | Vision Encoder | $CHAIR_s$↓ | $CHAIR_i$↓ | Coverage↑ | Avg. Length↑ | Avg. Object↑ |
|---|---|---|---|---|---|---|
| CCEval (Ours) | CLIP-L-224x | 79.00 | 21.70 | 32.04 | 110.36 | 9.12 |
| | CLIP-L-336x | 79.00 | 18.90 | 36.00 | 111.55 | 9.19 |
| | CLIP-L-224x (SW) | 72.00 | 18.70 | 36.89 | 110.43 | 8.65 |

window approach for efficiency, as detailed in the Appendix. Table 6 shows a constant decrease in hallucination and increase in object coverage. Additionally, we assess LLaVA with Llama $2_{13B}$, varying the resolution from 112x to 336x. For the 112x112 resolution, the original image was downscaled to 112x and subsequently upscaled to 224x before being input to CLIP-Large-224x. Table 5 gives a consistent observation that larger input resolution can reduce hallucination.

**Conclusion.** Through our systematic analysis of object existence hallucination, we summarize several insights: (1) Enlarging the language decoder does mitigate hallucination, but improvements are not huge. (2) Expanding the volume of instruction data actually increases hallucination. Upon inspection of training data, we find certain objects described in the captions might not be grounded by the vision encoder. (3) To validate our hypothesis in (2), we show improving input image resolution significantly reduces hallucination by enhancing model grounding ability.

Reflecting on these findings, we attempt to provide an explanation for the reason of object existence hallucination in detailed captions. The process of image captioning in LVLMs can be perceived as a form of information mapping or translation. Ideally, the goal is to have a direct one-to-one correspondence between objects identified in the image and those mentioned in the captions. Objects successfully grounded by the vision encoder form accurate correspondence, making this as **contextual knowledge** in the model, following (Neeman et al., 2023). When objects in training caption fail to ground by the vision encoder, the model learns **parametric knowledge**, the knowledge encoded in the model's parameter. This kind of knowledge is the association of objects in the language with other words instead of with corresponding image object feature. During inference, when the model draws from parametric knowledge, it attempts to "guess" details not grounded by the vision module and is perceived as object existence hallucination.

## 3 HALLUCINATION CONTROLLING

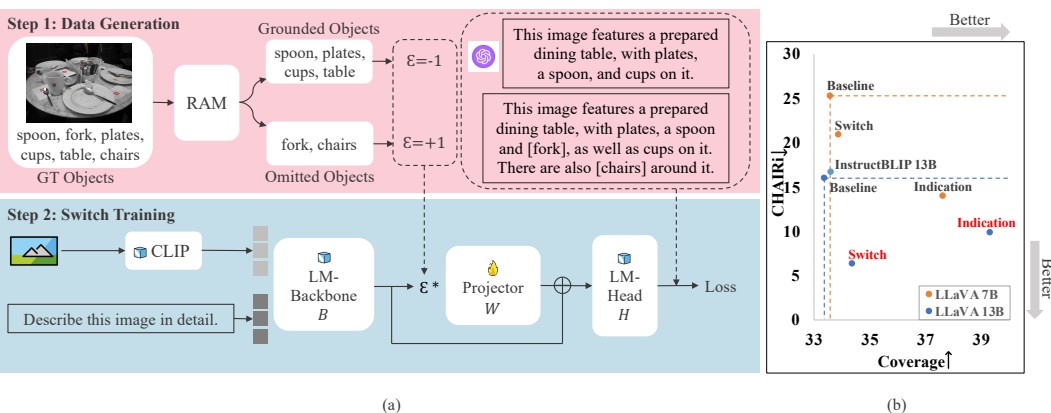

(a) (b)

Figure 2: (a) shows the overall training pipeline of HallE-Switch. When generating data, we use RAM to separate ground truth objects to visually grounded and omitted objects. Then, we utilize GPT-4 to convert this existing list of grounded objects into a caption as contextual data, and we assign $\varepsilon$ as $-1$. We put bracket around omitted objects in the original LLaVA caption as parametric joint data and assign $\varepsilon$ as $+1$. During training, we supervise using contextual only data and parametric joint data, pass in $\varepsilon$ as $-1$ or $+1$, respectively. (b) shows our methods consistently outperforms LLaVA baselines and InstructBLIP$_{13B}$. Indication is w/ ind. in Table 7, and Switch is -1 in Table 8.

Following our previous explanation that parametric knowledge leads to hallucination, we can eliminate the parametric knowledge and refrain the resulting model from guessing related objects. However, such guessing is necessary depending the details required by the downstream task, as shown in Appendix. Therefore, our approach work towards controlling and balancing parametric knowledge rather than eliminating.

We introduce *HallE-Switch*, a LVLM designed to control the extent of parametric knowledge within detailed captions. For this purpose, we developed two datasets: the first captures solely contextual knowledge, while the second merges both contextual and parametric knowledge. Using these datasets, we integrated a control projector into the model to control parametric knowledge.

## 3.1 Data Generation

**Grouping using RAM.** We begin by passing MSCOCO's ground truth objects to the open vocabulary detector RAM (Zhang et al., 2023b). RAM categorizes the objects into two groups: "grounded" (contextual group) and "omitted" (parametric group). This step aims to simulate the maximum visual granularity achievable by a vision encoder in LVLMs.

**Contextual Data Generation.** Our first dataset involves generating detailed captions using only objects from the contextual group. To do this, we feed MSCOCO source labels (including object classes, bounding boxes, and short captions) into GPT-4. We adhere to LLaVA's caption creation pipeline and provide prompt in Appendix.

**Parametric Joint Data Generation.** The second dataset incorporates both contextual and parametric knowledge. Here, we begin with LLaVA's original detailed captions and annotate objects from the parametric group with special tokens. Specifically, we enclose the "omitted" objects with brackets. Formally, if $S$ denotes the original image caption sentence and $X = \{x_1, ..., x_n\}$ represents a set of undetected objects, our data processing can be represented as:

$$S_{new} = \text{replace}(S, x_i, [x_i])$$

The purpose of bracketing the parametric objects is twofold: it serves as an indicator during inference and provides a hint during training.

## 3.2 Hallucination Switch

Inspired by LM-Switch (Han et al., 2023), we address hallucination by adding a control parameter $\varepsilon$ that serves as a "switching value", e.g., $+1$ for permitted imagination and $-1$ for restricting imagination, as depicted in Figure 2(a). Let $M$ represent the LLM with fixed parameters: $M(x) = H(e_v)$ where $H$ stands for LM-head and $e_v = B(x)$ is the output word embedding from LM-backbone. We modify $M' = M(\varepsilon W)$, thus making output word embedding $e'_v = e_v + \varepsilon W e_v$, leading to the derived model $M'$ as:

$$M'(x) = H(B(x) + \varepsilon W(B(x))).$$

The learned projector $W$ can be regarded as the transformation from a generic word space to the object sensitive word space, where word-word semantic correspondence is optimized to object correspondence, and $\varepsilon$ governs the intensity of imagining related objects.

**Training.** To train such a controlling parameter, we leverage the contrastive training data covering both contextual and parametric datasets in Sec 3.1. For data with only contextual knowledge, we assign $\varepsilon = -1$ when inputting into the model. In contrast, for data with both contextual and parametric knowledge, we use $\varepsilon = 1$. Notably, only the linear layer $W$ is fine-tuning throughout the training phase.

**Inference.** At the inference stage, $\varepsilon$ can adopt any value within the interval $[-1, 1]$. Specifically, an $\varepsilon$ value of $-1$ corresponds to minimal reliance on parametric knowledge, whereas a value of $1$ indicates a strong inclination towards such knowledge. Detailed theoretical explanation on why HallE-Switch works are elaborated upon in the Appendix.

# 4 Experiment

## 4.1 Finetune on Parametric Joint Data

Before we present experiments on HallE-Switch, we show the upper-bound experiment results on how well the model can indicate parametric knowledge. We directly finetune the LLaVA model on parametric joint data. Intuitively, the model is trained on data indicating parametric knowledge. Its output should identify parametric knowledge accurately. Specifically, the model should put a bracket around every "guessed" objects for indication of hallucination.

Therefore, we evaluate the object hallucination in three different settings: 1. Evaluation only on indicated objects: We do CCEval only on objects inside the bracket. The result should reflect a high level of hallucination. 2. Evaluation without indicated objects: We disregard objects in bracket and calculate CCEval. The result should reflect a low level of hallucination. 3. Evaluation with indicated

Table 7: Comparison between baselines and the effect of indication on *CCEval*. 'Only ind' means evaluation only on indicated objects; 'w/o ind' means evaluation without indicated objects; 'w/ ind' means evaluation with indicated objects.

| Setting | LLM | Resolution | $CHAIR_s\downarrow$ | $CHAIR_i\downarrow$ | Coverage↑ | Avg. Length↑ | Avg. Object↑ |
|---|---|---|---|---|---|---|---|
| 158K baseline | LLaVA$_{7B}$ | 224x | 82.00 | 25.30 | 33.58 | 109.89 | 9.31 |
| only ind. | LLaVA$_{7B}$ | 224x | 53.00 | 63.90 | 12.01 | – | 1.66 |
| w/o ind. | LLaVA$_{7B}$ | 224x | 57.00 | 17.10 | 37.60 | 108.94 | 7.63 |
| w/ ind. | LLaVA$_{7B}$ | 224x | 57.00 | 14.00 | 37.62 | 108.94 | 9.22 |
| 158K baseline | LLaVA Llama 2$_{13B}$ | 336x | 64.00 | 16.00 | 33.37 | 108.52 | 8.92 |
| only ind. | LLaVA Llama 2$_{13B}$ | 336x | 52.00 | 62.31 | 19.90 | – | 1.3 |
| w/o ind. | LLaVA Llama 2$_{13B}$ | 336x | 52.00 | 11.62 | 34.70 | 106.94 | 7.23 |
| w/ ind. | LLaVA Llama 2$_{13B}$ | 336x | 52.00 | 9.86 | 39.31 | 106.94 | 8.52 |

Table 8: Performance of *HallE-Switch*.

| Switch | LLM | Resolution | $CHAIR_s\downarrow$ | $CHAIR_i\downarrow$ | Coverage↑ | Avg. Length↑ | Avg. Object↑ |
|---|---|---|---|---|---|---|---|
| 1 | LLaVA$_{7B}$ | 224x | 89.00 | 26.60 | 32.80 | 108.54 | 9.72 |
| 0.5 | LLaVA$_{7B}$ | 224x | 85.00 | 27.92 | 34.02 | 109.33 | 8.81 |
| -0.5 | LLaVA$_{7B}$ | 224x | 81.00 | 24.88 | 35.87 | 118.08 | 8.04 |
| -1 | LLaVA$_{7B}$ | 224x | 76.00 | 20.90 | 33.88 | 133.79 | 8.02 |
| 1 | LLaVA Llama 2$_{13B}$ | 336x | 65.00 | 14.58 | 36.14 | 102.18 | 8.37 |
| 0.5 | LLaVA Llama 2$_{13B}$ | 336x | 65.00 | 14.44 | 32.32 | 103.51 | 8.45 |
| -0.5 | LLaVA Llama 2$_{13B}$ | 336x | 66.00 | 13.79 | 33.07 | 105.57 | 8.41 |
| -1 | LLaVA Llama 2$_{13B}$ | 336x | 43.00 | 6.37 | 34.37 | 136.28 | 8.79 |

objects: We calculate CCEval on all objects. Due to modification of the settings, we slightly change definition of CHAIR scores in CCEval as detailed in Appendix.

**Evaluation only on indicated objects.** The hallucination level for indicated objects, denoted as $CHAIR_i$, is 63.90 for LLaVA$_{7B}$ and 62.31 for LLaVA$_{13B}$. It is considerably higher than the baselines all other models. Concurrently, their coverage is 12.01 and 19.09 for LLaVA$_{7B}$ and LLaVA$_{13B}$, respectively, which both of them significantly lower than the coverage of 33.58 for LLaVA$_{7B}$ baseline model. The experiment results show the object within special tokens has significantly higher hallucination rate and lower coverage rate which support our assumption that the object inside the indication tokens are objects from parametric knowledge.

**Evaluation without indicated objects.** For objects outside of the special token scope, we found that hallucination is markedly reduced, CHAIR$_s$ decreased from 82 to 57 which is 30.5% percent improvement and CHAIR$_i$ decrease 32.4%, from 25.3 to 17.10, compared to the baseline. This suggests that the model is less prone to make erroneous assumptions for objects not marked by brackets. This is interesting because the model perfectly capture the intention of marking parametric objects in training data and replicate the behavior during inference.

**Evaluation with indicated objects.**: We observe a significant decline in the hallucination without any reduce in object coverage. LLaVA$_{7B}$ CHAIR$_i$ improved from 25.30 to 14.00 which has 44.66% improvements. For LLaVA$_{13B}$ CHAIR$_i$ improved from 16 to 9.86 which also has 38.38% improvements.

## 4.2 HALLUCINATION CONTROLLING

During model inference, we select 4 different $\varepsilon$ , ranging from $-1$ to $+1$. As shown in Table 8, we evaluate HallE-Switch$_{7B}$ and HallE-Switch$_{13B}$ model, which use LLaVA as backbone. For the 7B model, we train it by removing the special token in parametric joint data, showing that indication is not a necessary condition for switch to work. $\varepsilon = -1$ means the switch trained purely on contextual only data, which try to minimize the hallucination, where $\varepsilon = +1$ switch to maximize parametric knowledge output. The results show that as $\varepsilon$ increase, CHAIR$_i$ increase from 20.90 to 26.6, the coverage keeps at a similar level.

For the 13B model, we keep the indication inside the parametric joint data. The HallE-Switch$_{13B}$ achieves the best in object existence hallucination metric. With switch set to -1 and indication, we have CHAIR$_s$ = 43 versus baseline's 64 and CHAIR$_i$ = 6.37 vs baseline's 16, and the coverage of the model is not decreasing.

## 5 RELATED WORK

**Large Vision-Language Models (LVLMs).** The rapid advancements in Large Language Models (LLMs) (Touvron et al., 2023a; Chung et al., 2022; Touvron et al., 2023b; Anil et al., 2023; Driess et al., 2023; Scao et al., 2022; OpenAI, 2023) combined with a surge in open-source initiatives, have paved the way for the emergence of extensive vision-language models (Liu et al., 2023d; Goyal et al., 2017; Zhu et al., 2023; Sun et al., 2023; Ye et al., 2023; Bai et al., 2023; Chen et al., 2023; Peng et al., 2023). LLaVA introduced the concept of integrating a simple projector during LLM fine-tuning. Chatspot (Zhao et al., 2023b) follow LLaVA's model structure, but embed region of interest into instruction data. GPT4RoI (Zhang et al., 2023a) and Shikra (Chen et al., 2023) add grounding tasks to LLaVA structure models, and achieve great performance on various tasks. Instead of using detector to provide region information to the model, we use detector to filter objects for alignment between vision and language information. Concurrently, BLIP2 (Li et al., 2023d) and InstructBLIP (Dai et al., 2023) presented Q-former-based LVLMs. Multimodal-GPT (Gong et al., 2023) and Otter (Li et al., 2023b) aims to improve OpenFlamingo's (Alayrac et al.; Awadalla et al., 2023) directive adherence. mPLUG-Owl (Ye et al., 2023) suggests a two-step method: first train vision models, then refining the language model using techniques like LoRA. Our work utilize a linear layer to control object existence hallucination within LVLMs.

**Evaluation on LVLMs.** The evaluation of large vision-and-language models (LVLMs) (Yu et al., 2023; Liu et al., 2023b;c) is notably challenging due to the intricate nature of generation tasks they undertake. Some of the VQA-based benchmarks (Antol et al., 2015; Hudson & Manning, 2019; Gurari et al., 2018) require models to identify objects, colors, or quantities, while others (Liu et al., 2023e; Li et al., 2023c; Lu et al., 2022) offer multiple-choice questions. POPE (Li et al., 2023e) and MME (Fu et al., 2023) include object hallucination evaluation like paired yes/no questions on object existence, color, counting, OCR, and etc. While VQA-based benchmarks are cheap and straight-forward, we find them cannot accurately reflect object hallucination for detailed captions. Besides VQA benchmarks, ROUGE (Lin, 2004; Elliott & Keller, 2014) use n-gram to evaluate similarity between ground truth and model inferences. CIDr (Vedantam et al., 2015) is a triplet-based method of collecting human annotations to measure consensus. CHAIR (Rohrbach et al., 2018) evaluate caption hallucination based on object concept. These methods are constraint by ground truth length or word variance and cannot clearly reflect hallucination with object coverage information. Wang et al. (2023) try use a language model to predict whether the caption exist hallucination, which is cheaper than GPT-4. Our work introduces CCEval, including CHAIR metrics, object coverage, average sentence length and number of objects to overcome limitations of previous evaluations.

**Hallucination** Hallucinations (Ji et al., 2023a; Shi et al., 2023; Lin et al., 2021) have been widely studied in traditional NLG (Ji et al., 2023b) tasks, including machine translation (Zhou et al., 2020; Lee et al., 2019), data-to-text (Rebuffel et al., 2021; Kasner & Dušek, 2022; Lee et al., 2022), summarization (Cao et al., 2022), dialogue (Dziri et al., 2022) and QA (Shuster et al., 2021). For LVLMs, previous studies have been mainly focusing on object hallucination (Marino et al., 2019; MacLeod et al., 2017; Li et al., 2023a;e). POPE (Li et al., 2023e) reveals object existence hallucination may related with label distributions, such as object co-occurance. Earlier than POPE, Biten et al. (2022) balance object co-occurance to decrease hallucination. LRV (Liu et al., 2023a) finds the cause of hallucination in VQA benchmarks, especially unbalanced answer distribution and lack of negation information. Our work raise another important cause: misalignment between the vision and language information captured by models. More interestingly, we can explain balancing labels in Biten et al. (2022) as trying to weaken the parametric knowledge caused by the misalignment.

## 6 CONCLUSION

In summary, this study delves deep into the object hallucination phenomena within the detailed captions of LVLMs, advancing understanding of the accuracy and unwarranted inference in describing visual details. We introduce a novel and comprehensive evaluation method for object existence hallucination in detailed captions. We conduct an in-depth and component-wise analysis of LVLMs, meticulously examining each element that might result in hallucination. We further identify an alignment issue between the vision encoder and the instruction data. To alleviate such hallucination, we introduce controlling parameters over LVLMs to condition the inference of objects.

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
