# HallE-Switch: Rethinking and Controlling Object Existence Hallucinations in Large Vision-Language Models for Detailed Caption

## A  Appendix

### A.1  Why not all Hallucination Bad?

In our study, we define "object existence hallucination" to be a phenomenon where a description makes reference to objects that is not present in the image. However, such hallucinations, when properly harnessed, can be regarded as instances of imagination. Human beings frequently use imagination to successfully accomplish tasks, often without even realizing it. Here, we present several scenarios in Table 1, Table 2, and Table 3 in which imagination and judicious inference prove to be beneficial for various downstream applications, including robotic manipulation and content moderation, among others. Our work successfully exercise control over the extent of generalization rather than attempting to eliminate all instances of imagined objects. More importantly, our model can indicate imagined objects with `[object]` marker.

---

**First Case:** In the context of robotic applications, Large Vision-Language Models (LVLMs) encounter challenges when camera does not capture all objects needed. In such situations, it becomes crucial for a detailed caption to infer the surrounding objects. For instance, in this scenario, if the robot is ask for manipulating the chairs, it can successfully locate the chairs and notice people sitting on the chairs, even if they are not shown in the picture.

**Image:**

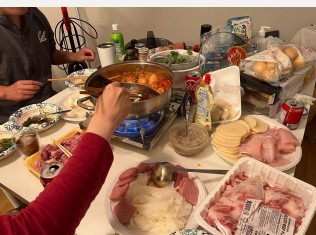

**Caption:**
The image depicts a group of people gathered around a dining table, preparing a meal together. The table is filled with various food items, including several bowls containing different dishes, and a few plates with different types of food. There are also multiple bottles and cups scattered across the table, likely containing beverages or condiments. Two people can be seen in the scene, one on the left side and another on the right side of the table. They are actively engaged in the cooking process, with one person stirring a pot of food on the stove. The table is surrounded by chairs, with one on the left side and another on the right side. In addition to the food and people, there are a few utensils on the table, such as a spoon and a knife. A cell phone can also be seen on the table, possibly being used to look up recipes or communicate with others.

---

Table 1: Beneficial hallucination in robotic applications. Caption is from LLaVA1.5. Imagined objects are highlighted.

**Second Case:** In the context of content moderation, Large Vision-Language Models (LVLMs) need to infer objects to recognize cases of violation. For instance, in this scenario, a man is not explicitly present in the image but can be correctly inferred to help moderation.

**Image:**

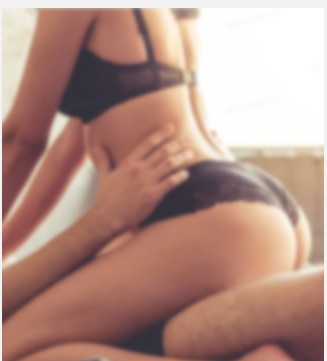

**Caption:**

The image features a `man` and a woman lying on a bed, with the woman sitting on top of the man. The woman is wearing black lingerie, and the man has his hands on her hips, possibly holding her in place. The couple appears to be enjoying a moment of intimacy together. The bed occupies most of the image, with the couple being the main focus. The woman's black lingerie is prominently visible, while the man's hands are placed on her hips, emphasizing the close and intimate nature of their interaction.

Table 2: Beneficial hallucination in content moderation. Caption is from LLaVA1.5. Imagined objects are `highlighted`.

**Third Case:** In the context of robotic applications, Large Vision-Language Models (LVLMs) encounter challenges when dealing with poorly captured images. These images often feature partially occluded or blurred objects. In such situations, it becomes crucial for a detailed caption to go beyond what is visible and accurately infer the presence of occluded objects. This capability is essential to address safety considerations. For instance, in this case, there is a cat behind the door reaching out its paw to play with the black ribbon, but the body of the cat is not explicitly shown. If the model can successfully recognize a cat, the robot should refrain from pushing the door, thus ensuring the safety of both the robot and the cat.

**Image:**

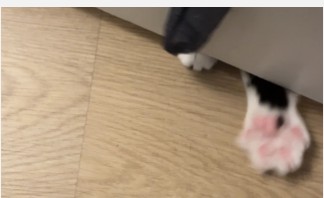

**Caption:**

The image features a black and white `cat` with pink paws, standing on a wooden floor and looking under a door. The cat appears to be curious about what is happening behind the door. The door is positioned on the left side of the image, taking up a significant portion of the background. The cat is located towards the right side of the image, with its paws visible as it peers under the door.

Table 3: Beneficial hallucination in robotic applications. Caption is from LLaVA1.5. Imagined objects are `highlighted`.

## A.2 MORE DETAILS ON ANALYZED MODELS

In Section 2, we conduct analysis on different LVLMs. Here, we provide more details of each model:

**LLaVA** uses a linear projector to map visual token as a soft-prompt into LLM input tokens. LLaVA has a two-stage training, where the initial stage focuses on simple caption pretraining solely for the linear projector, while the subsequent stage finetunes both the projector and LLM on instruction data. Instruction data leverages language-only GPT-4 by inputting visual ground truth from COCO dataset.

**InstructBLIP** adopts the BLIP-2 architecture, and is distinguished by its training of a Q-former, which bridges the frozen vision encoder and LLM. InstructBLIP's instruction fine-tuning spans across 26 distinct datasets.

**Shikra** mirrors LLaVA's model structure. It eliminates the pretrain stage, but introduce grounding task during finetuning. Shikra is trained on multiple datasets like InstructBLIP.

## A.3 TRAINING DATA QUALITY

We sampled three images from the MSCOCO dataset, as illustrated in Table 4. For each image, we present the visual content, a detailed caption generated by GPT-4 based on bounding boxes and regional captions, and the object ground truth labels derived from the MSCOCO dataset annotations.

**First Image.** This image showcases three remote controls, posing a unique challenge. The ambiguity lies in distinguishing the type of remote, be it for video games or televisions. Additionally, the remotes near the wall are relatively small, making them harder to see. A person is partially visible in the image, with only their knees being evident. There is also an incomplete bottle on the image's left side.

**Second Image.** A significant issue with this image is the individuals visible behind a window. Not only detection models struggle to recognize them, even for human observers, counting the individuals inside the train is challenging. This particular issue is prevalent in many of MSCOCO's traffic-related images.

**Third Image.** This image depicts a table around which two individuals are seated. Close observation is required to recognize both individuals. A clear indication of one person is a pair of hands, while the other individual is considerably harder to spot. Additionally, there seems to be an annotation error in the ground truth labels: it indicates only one bowl, neglecting to include plates and other items. Upon closer inspection, there are four plates.

## A.4 PROMPT

Table 5 is the prompt we use for caption object extraction.

Table 6 is the prompt we use for hallucination object matching.

Table 7 is the prompt we use for finding ground truth object coverage.

## A.5 SLIDING WINDOW

We pad the original image to dimensions of $672 \times 672$. Then, we divide the image into a $3 \times 3$ grid, where each cell measures $224 \times 224$. The encoder processes these cells sequentially, starting from the top-left and moving towards the bottom-right. The visual tokens from each cell are then concatenated.

## A.6 EXPERIMENT SETTINGS

The LLaVA we used in all experiments is pretrained on on LCS-558k which is subset of LAION (Schuhmann et al., 2021), CC (Sharma et al., 2018) and SBU (Ordonez et al., 2011) data, and finetuned on Instruct-158K instruction data. We use Vicuna version 1.3 as initialized language decoder and CLIP-Large as vision encoder. The 158K finetuning data consists of detailed caption, complex reasoning, and conversation.

We employed the RAM detector, specifically the RAM-14M variant, which uses a Swin-Large backbone. In the data generation stage, we focused on the 'detailed caption 23K' file from the LLaVA Instruction set comprising 158K entries. This file was generated using a specific prompt provided by LLaVA repository[1]. Our preprocessing involved adding brackets '[]' around objects filtered by the RAM detector to uniquely identify them in the captions.

---

[1] `https://github.com/haotian-liu/LLaVA/tree/main/playground/data/prompts/detail_description`

| Image | Caption | GT Labels |
|---|---|---|
| 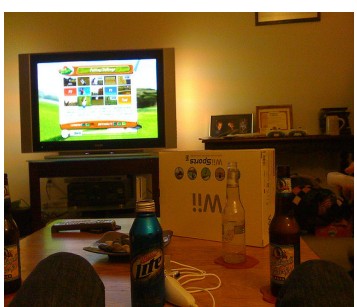 | The image depicts a cozy entertainment room featuring a flat screen TV situated against the wall. A dining table can be seen next to the TV, with some bottles and a cup placed on top of it. Multiple video game controllers, including Wii remotes, are also present on the dining table, suggesting that people are using the TV to play video games. To the far right of the scene, another remote is located close to the wall. A person appears to be sitting or standing behind the dining table, likely either watching the game or waiting for their turn to play. The assortment of bottles on the table suggests that the guests are enjoying drinks during their video game session. | tv: 1 cup: 1 remote: 3 bottle: 4 person: 1 dining table: 1 |
| 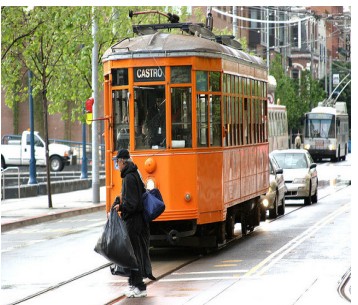 | In the image, an orange mass transit trolley is making its way through a city. A person is crossing the street in front of the trolley while holding a garbage bag, appearing to be cautious about the approaching vehicle. Another person is standing close to the person crossing the street, and there are two more individuals nearby. In the scene, various vehicles surround the trolley, including cars, buses, and trucks. One of the cars is parked right behind the trolley, while another is situated farther back. Two buses can be seen, with one staying behind the trolley and the other on the right side of it. A truck is also present at the far left side of the scene. A woman nearby is holding a handbag, completing the busy urban setting. | car: 3 bus: 2 truck: 1 train: 1 person: 5 handbag: 1 |
| 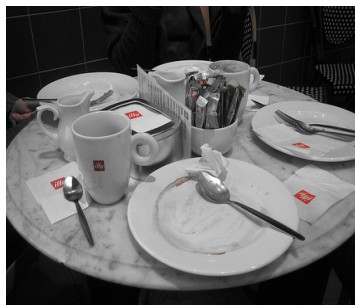 | The image shows a table in a restaurant, which appears to have been recently used for a meal. The table is set with four place settings, including white dishes. On the table, there are multiple cups, a bowl, and a mix of silverware like forks and spoons laid out. Some of the dishes look dirty, with used napkins and eating utensils scattered around. In the background, two people can be seen, one situated on the left side and another on the right side behind the table. There are also chairs located near the table, with one chair positioned close to the left side and another chair closer to the right side of the frame. | cup: 5 spoon: 3 fork: 2 person: 2 chair: 2 bowl: 1 dining table: 1 |

Table 4: Quality of training data.

Hallucination switch only involves only finetuning a linear layer added before $lm\_head$ layer. We freeze all other layers and only finetune switch layer with 3 epochs. The dataset is generated data

with associated $\varepsilon$ value. We have 10K contextual only detailed caption data and 23K parametric joint detailed caption data.

## A.7 MODIFIED CCEVAL OF SECTION 4.1

1. Evaluation only on indicated objects

$$\text{CHAIR}_i = \frac{|\{\text{hallucinated objects\_w/\_indication}\}|}{|\{\text{all objects w/\_indication mentioned}\}|}$$

$$\text{CHAIR}_s = \frac{|\{\text{sentences with hallucinated object\_w/\_indication}\}|}{|\{\text{all sentences w/\_indication}|}$$

2. Evaluation without indicated objects

$$\text{CHAIR}_i = \frac{|\{\text{hallucinated objects\_w/o\_indication}\}|}{|\{\text{all objects w/o\_indication mentioned}\}|}$$

$$\text{CHAIR}_s = \frac{|\{\text{sentences with hallucinated object\_w/o\_indication}\}|}{|\{\text{all sentences w/o\_indication}|}$$

3. Evaluation with indicated objects

$$\text{CHAIR}_i = \frac{|\{\text{hallucinated objects\_w/o\_indication}\}|}{|\{\text{all objects mentioned}\}|}$$

$$\text{CHAIR}_s = \frac{|\{\text{sentences with hallucinated object\_w/o\_indication}\}|}{|\{\text{all sentences}\}|}$$

## A.8 QUALITATIVE RESULTS OF HALLE-SWITCH

This section shows qualitative results for HallE-Switch in Table 8, Table 9, Table 10, Table 11, and Table 12. $\varepsilon = -1$ means no imagination and $\varepsilon = 1$ means max imagination. Both models can output indication.

## A.9 THEORETICAL EXPLANATIONS

**HallE-Switch Formulation**

HallE-Switch follows LLaVA's training strategy, which freezes vision encoder and language model, only finetune the projector for the first stage. The visual encoder $g(\cdot)$ transfer image $X_v$ to visual features:

$$Z_v = g(X_v)$$

The projector $W$ connects image feature to the word embedding space:

$$H_v = W \cdot Z_v$$

.

In the second stage, both projector and language model is trained. The model is trained on two type of data: 1. Contextual only data. 2. Contextual and parametric combined data.

Language model trained on contextual only data has a distribution initiating from $\pi$. The other language model trained on combined data has a distribution initiating from $\pi'$

With the Theorem 1 in LM-Switch (Han et al., 2023), under the same assumptions, there exists an matrix $W$, transforming a word embedding $E$ to $WE$, which is equivalent to let a LM simulate the text distribution initiating from another distribution.

Inspired by the Theorem, we propose a linear transform in word embedding space for LVLM. Let $M$ be the finetuned LLM, with switch layer/projector denoted as $W$, we replace each word embedding $e_v$ with $e_v + \varepsilon W e_v$, making the new language model $M' = M(\varepsilon W)$ as HallE-Switch's language

decoder. $\varepsilon$ is adjustable from -1 to 1. We assign $\varepsilon = -1$ and fit contextual only data; assign $\varepsilon = 1$ with the contextual and parametric combined data. After finetuning the HallE-Switch, the user only needs to specify a switch value $\varepsilon \in [-1, 1]$ and do normal vision language task like image captions. We use maximal likelihood as the training objective.

**Continuous Control**

The design of LM-Switch maintains a linearity guarantee, with proof of the switch model's distribution is close to a linear interpolation. Let $\lambda_{max}$ be the maximum eigen-value of W. When varying $\varepsilon'$s value,

$$\|P(\cdot|k\varepsilon, W) - (P(\cdot)(1-k) + kP(\cdot|\varepsilon, W))\|_1 \le 2|k(1-k)|\varepsilon^2 L^2 \lambda_{max}(e^{\lambda_{max}} - 1)$$

distribution of the switch model is close a linear interpolation of $M$ and $M'$, meaning that the model distribution changes linearly. Therefore, our method can take any $\varepsilon$ between 1 and -1.

For HallE-Switch, the core idea is: Assuming LLM is good enough to represent an equivalent distribution with HMM; there exist an matrix W, so that after transferring word embedding $E$ to $WE$, the LLM's originally simulate the text distribution starting with initial state $\pi$ will turn to be equivalent to a distribution starting with initial state $\pi'$.

According to experiments in Section 4.1, it reveals language models can distinguish inference object and observed objects by adding special tokens. Because, decoder based language models generate sequences in an auto-regressive way. We followed the proof in Han et al. (2023): if we assume **O** as our observation space which means a set of m observations. We assume $\mathbf{c}(o_1, ..., o_{t-1})$ as a contextual vector, $\mathbf{E} = (e_o, ...) \in \mathbb{R}^{\times|\mathcal{O}|}$ as word embedding. We can represent word logits as $\mathbf{l} = \mathbf{c}(o_1, ..., o_{t-1})^\top \mathbf{E}$. In attention mechanic, it will pass through a softmax operator to get the distribution over words, We use the same assumption in LM-Switch method to assume a linear formulation and let the conditional probability in language model $P(o_t|o_1, ..., o_{t-1}) = \mathbf{c}(o_1, ..., o_{t-1})^\top e_{o_t}$. We can get the full probability by using chain-rule: $\prod_{t=1}^T P(o_t|o1, ..., o_{t-1}) = P(0_1, ...o_{t-1})$. We assume our LLMs are good enough to represent an equivalent distribution with HMM and full column-rank for $\mathbf{E}$, $\mathbf{p}(o)$.

Our conclusion is: Applying a linear transformation on word embedding space is equivalent to a shift from one initial condition to another. This is the reason that we want to shift a language model with distribution produce higher inference imagination conditional probability to a lower one.

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

**Caption Object Extract Prompt:**

**User:**
I have a description of an image, and I want to get objects from this description and return these objects in a list the object should be a noun, and I don't want duplicated objects. I don't want the scene name to be included, such as some caption describing the image as a scene or depicting a position or a situation or place, this thing is not an object, and doesn't need to be included. Here some objects are inside [] which we want to ignore. Here are some examples:
**Example 1:**
**Input:**
caption = "The image features a bathroom sink situated under a large mirror. The sink is accompanied by a soap dispenser, and there are multiple toothbrushes placed around it. A few cups can be seen scattered around the sink area as well. \n \n In addition to the sink, there is a toilet visible to the left side of the bathroom. The overall scene gives an impression of a well-equipped and functional bathroom space. Also a [brush] can been seen."
**Answer:**
objects = ['sink', 'mirror', 'soap dispenser', 'toothbrush', 'cup', 'toilet']
Here we can see [brush] is ignored because its inside []. bathroom is the place not object, so not included.

**Example 2:**
**Input:**
caption = "The image depicts a cluttered dining room with a large kitchen table in the center. The table is covered with dirty dishes, including plates, bowls, cups, and utensils. There are several chairs around the table, with some placed closer to the center and others positioned at the edges. In addition to the dishes, there is an apple sitting on the table, likely left over from a meal or snack. A bottle of water can be seen on the table as well, and a [flower], adding to the messy atmosphere of the room."

**Answer:**
objects = ['table', 'dish', 'bowl', 'cup', 'utensil', 'chair', 'apple', 'water']

Here [flower] is in [], should be ignored. Here dining room and room are places, so ignored, not in objects.

**Example 3:**
**Input:**
caption = "The image depicts a busy city street with a pedestrian crossing in a sunny day. A man is walking across the street, carrying a backpack and wearing a jacket."

**Answer:**
objects = ['street', 'pedestrian crossing', 'man', 'backpack', 'jacket']

Here 'city' is a place, so not an object so not included in objects. 'The image depicts' is about the image caption task, so not an object in the scene. 'sunny' or 'sunny day' or 'day' are not objects in the image, this is a time situation so not an object, can't in objects.

**Example 4:**
**Input:**
caption = "The image depicts an office cubicle with a desk in the center. The desk is equipped with a computer, a keyboard, and a mouse."
**Answer:**
objects = ['desk', 'computer', 'keyboard', 'mouse']

Here the office is a place so not in objects. Here 'center' is not object, 'center' is position, not an object, same thing like 'left' or 'right' etc.

**Inputs:**
caption = cap
**Answer:**
objects =

Table 5: Caption object extract prompt

**Hallucination Prompt:**

**User:**
I have two lists of objects, list_A, and list_B, I want to return a list hallucination which finds items in list_B don't appear in list_A, sometimes same object can be expressed in different ways in list_A and list_B, we treat different expression but similar meaning objects as matched, not include in mismatch list.

**Example 1:**
**Input:**
list_A = ['reflection of light', 'view of office building', 'street chair', 'white car', 'red car', 'dark hair', 'bagpack', 'black shoes', 'dark pants', 'bikes', 'street', 'street light']
list_B = ['two cars', 'dark bagpack', 'yellow jacket', 'light', 'brick building', 'wood chair', 'chair', 'green car', 'dining room table', 'bike', 'city street', 'traffic light', 'sedan']
**Answer:**
In this example, 'two cars' is just object 'car', we don't care about the number of object. Although 'bikes' and 'bike' is not the same word, but we treat singular nouns and plural nouns as the same thing, so it's not mismatch. Here in list_A's 'street' and list_B's 'city street' are not exactly match but actually, city street can been seen as a kind of street, since city street is still a street, just in city, so they are similar meaning, we don't treat it as a mismatch, even 'city street' seems more specific, but we only still treat it as a match not hallucination. Although there is 'street light' in list_A but 'traffic light' is a different object, 'light' and 'street light' are aiming for providing lights, but 'traffic light''s purpose is providing signal, so they are different object. 'Sedan' is a different kind of car, so 'sedan' match 'car'.

hallucination = ['yellow jacket', 'dining room table', 'traffic light']

**Example 2:**
**Input:**
list_A = ['bag', 'cloth', 'boy', 'Drinking glasses', 'table']
list_B =['backpack', 'jacket', 'young man', 'cup', 'kitchen table']

**Answer:**
In this example, 'bag' in list_A and 'backpack' in list_B have similar meaning, 'jacket' in list_B can be seen as a kind of 'cloth' in list_A still matching, and 'Drinking glasses' is kind of cup. In list_B 'kitchen table' is a kind of table as 'table' in list_A so there is no hallucination.

hallucination = []

**Example 3:**
**Input:** list_A = ['keyboard', 'mouse', 'moniter', 'cpu']
list_B = ['computer']

**Answer:**
Based on the objects, 'keyboard', 'mouse', 'moniter', 'cpu' they are all parts of a computer and they all appreared in list_A, and list_B's 'computer' is just a summary of all these objects, so there is no hallucination.

hallucination = []

**Inputs:**
list_A = {gt}
list_B = {cap_obj}
**Answer:**
hallucination =

Table 6: Object hallucination matching prompt

**Coverage Prompt:**

**User:**
I have two list of objects, list_A and list_B, I want to return a list named uncover which find items in list_B doesn't appear in list_A, sometimes same object can be expressed in different ways in list_A and list_B, we treat different expression but similar meaning objects as matched, not include in mismatch list.
**Example 1:**
**Input:**
list_A = ['two cars', 'dark bagpack', 'yellow jacket', 'light', 'brick building', 'wood chair', 'chair', 'green car', 'dining room table', 'bike', 'city street', 'traffic light', 'sedan'] list_B = ['reflection of light', 'view of office building', 'street chair', 'white car', 'red car', 'dark hair'] **Answer:**
uncover = ['reflection of light', 'dark hair']

In this example
'reflection of light' cannot find matched object in list_A, especially, 'light' is not equal to 'reflection of light'.
'view of office building' in list_B can find matched object 'brick building' although they are not exactly same but they point to similar object.
'street chair' in list_B can find 'chair', 'wood chair' in list_A which is an alternate expression of 'chair'.
'white car' in list_B can find 'two cars' in list_A.
'red car' in list_B can find 'two cars' in list_A.
'dark hair' in list_B cannot find anything similar in list_A

**Example 2:**
**Input:**
list_A = ['bag', 'cloth', 'boy', 'Drinking glasses', 'table']
list_B =['backpack', 'jacket', 'young man', 'cup', 'kitchen table', 'plate', 'apple']

**Answer:**
uncover = ['plate', 'apple']
In this example,
'backpack' in list_B can find 'bag' in list_A has similar meaning, matched.
'jacket' in list_B can be seen as a kind of 'cloth' in list_A still matching;
'young man' in list_B can match 'boy' in list_A;
'cup' in list_B is similar to 'Drinking glasses' in list_A;
'kitchen table' is a kind of table as 'table' in list_A so there is no uncovered items.
'plate' in list_B but no object has same or similar meaning in list_A.
'apple' in list_B but no object has same or similar meaning in list_A.

**Example 3:**
**Input:** list_A = ['keyboard', 'mouse', 'moniter', 'cpu']
list_B = ['computer']

**Answer:**
uncover = [] 'computer' in list_B can find 'keyboard', 'mouse', 'moniter', 'cpu' as whole thing in list_A, matched.

**Inputs:**
list_A = {cap_obj}
list_B = {gt}
**Answer:**
hallucination =

Table 7: Object coverage prompt

**First Case**

**Image:**

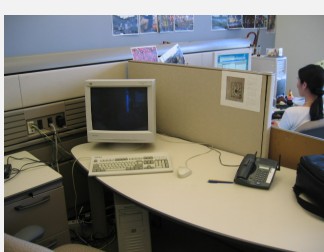

**Caption with $\varepsilon = -1$:**
The image features an office cubicle with a white desk and a computer setup. On the desk, there is a desktop computer with a monitor, keyboard, and mouse. A telephone is also present, indicating that this is a workstation with multiple functions. In addition to the main equipment, there are several books scattered around the workspace, suggesting that the user may be engaged in reading or research. A chair is positioned near the desk, ready for use, and a TV can be seen in the background, possibly for entertainment or information purposes during breaks.

**Caption with $\varepsilon = 1$:**
The scene is set in an office environment, with a computer [mouse] and keyboard prominently placed on a desk. The keyboard is positioned towards the left side of the desk, while the [mouse] is located more towards the right. A computer monitor is situated on the desk, slightly behind the keyboard, indicating a typical workstation setup. A telephone is also present on the desk, located towards the right side, suggesting that the workspace is used for communication and computer-related tasks. In addition to these items, there are two [cups] on the desk, one slightly to the left of the keyboard and the other a bit further to the right. These [cups] might be used for holding beverages or storing stationery. The office is furnished with two chairs, one on the left side of the desk and the other on the right. Both chairs are positioned to face the desk, indicating that they are used for seating while working or engaging in phone conversations. The scene also features two [tv] screens, one on the left side of the room and the other on the right. These [tv] screens might be used for displaying relevant information or providing entertainment during work breaks.

Table 8: Example captions generated with HallE-Switch$_{13B}$.

**Second Case**

**Image:**

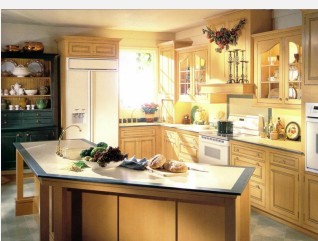

**Caption with $\varepsilon = -1$:**
The image depicts a large, clean, and well-organized kitchen with wooden cabinets and white countertops. The kitchen features a center island with various items placed on top of it, such as a knife, a loaf of bread, and some vegetables. There are multiple [bottles] and cups scattered around the kitchen, as well as a vase on the counter. In addition to the island, the kitchen is equipped with a refrigerator, a microwave, and two ovens, ensuring that it is well-equipped for cooking and food preparation. The presence of several vases and potted plants adds a touch of greenery and decoration to the space, making it inviting and pleasant.

**Caption with $\varepsilon = 1$:**
The scene is a large, well-lit kitchen with a center island. The island is covered in various food items, including [oranges], [apples], [bottles], [cups], [spoons], [knives], [bowls], [bananas], [broccoli], [potted plants], [vases], [bread], [cake], [wine glasses], [potted herbs], [potted flowers], [books], a [clock], a [potted plant] on the counter, a [vase] on the counter, a [bottle] on the counter, a [spoon] on the counter, a [knife] on the counter, a [bowl] on the counter, a [banana] on the counter, a [broccoli] on the counter, a [wine glass] on the counter, a [cup] on the counter, a [potted herb] in the background, a [potted flower] in the background, a [book] in the background, a [clock] in the background, a [potted plant] in the background, a [vase] in the background, a [bottle] in the background, a [spoon] in the background, a [knife] in the background, a [bowl] in the background, a [banana] in the background, a [broccoli] in the background, a [wine glass] in the background, a [cup] in the background. The kitchen is well-equipped with various appliances, including a refrigerator, oven, toaster, and microwave. The refrigerator, oven, and toaster are located along the wall, while the microwave is positioned above the counter. The kitchen also features a [sink] and ample counter space, making it a functional and inviting space for meal preparation and social gatherings.

Table 9: Example captions generated with HallE-Switch$_{13B}$.

**Third Case**

**Image:**

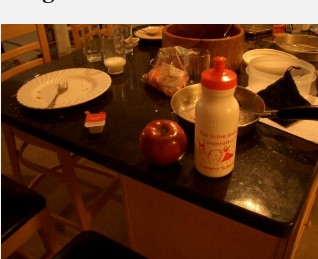

**Caption with $\varepsilon = -1$:**
The image features a kitchen counter with a few items on it. There is a bottle, an apple, and a bowl placed on the counter. The apple is situated near the middle of the counter, while the bottle is on the left side and the bowl is on the right side. In the background, there are two chairs, one near the bottom left corner and the other near the top right corner of the image. Additionally, two cups can be seen in the scene, with one placed near the bottom right corner and the other near the top center. A dining table is also visible, occupying the majority of the background.

**Caption with $\varepsilon = 1$:**
The scene is set in a kitchen, where a granite countertop is cluttered with various items. There are two cups on the counter, one located towards the left side and the other towards the right. A bottle is also present on the counter, positioned towards the left side of the scene. A bowl is placed near the center of the countertop, slightly to the left. Interestingly, there is an apple and a fork on the counter as well, located towards the right side of the scene. Additionally, a stool is positioned in the background, slightly to the right of the countertop. The kitchen is equipped with two chairs, one on the far left and the other on the far right, suggesting a space for dining or socializing.

Table 10: Example captions generated with HallE-Switch$_{13B}$.

**Fourth Case**

**Image:**

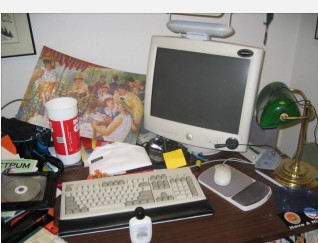

**Caption with $\varepsilon = -1$:**
The image features a computer desk with a desktop computer monitor placed on it. The monitor is surrounded by various items, including a keyboard positioned towards the front of the desk, a computer mouse located on the right side, and a [bottle] close to the monitor. There are also several cups placed on the desk, with one in the left corner, another in the middle, and the third one situated near the right side. Additionally, a pair of scissors can be found on the desk, lying horizontally between the monitor and the keyboard. The desk appears to be somewhat disorganized, giving the impression of a busy and active workspace.

**Caption with $\varepsilon = 1$:**
The scene is a cluttered desk with a desktop computer setup. The computer monitor is centrally located on the desk, with a keyboard placed directly in front of it. A mouse can be found to the right of the keyboard. Various objects are scattered across the desk, including a cup on the far left, a [spoon] slightly to the right of the cup, and a [potted plant] positioned towards the left side of the desk. There are also several books spread out on the desk, with one located towards the left, another towards the right, and two more towards the center. A [handbag] is placed on the desk, located towards the right side of the scene. Despite the clutter, the desk still has a functional workspace with all necessary computer peripherals and personal items.

Table 11: Example captions generated with HallE-Switch$_{13B}$.

**Fifth Case**

**Image:**

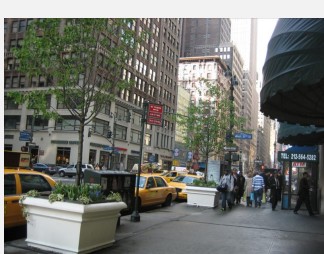

**Caption with $\varepsilon = -1$:**
"The image depicts a bustling city street with people walking down the sidewalk and various vehicles on the road. There are multiple cars, including a prominent yellow taxi cab, driving along the street. Additionally, there is a bus and a truck contributing to the busy traffic. Many pedestrians are present, some carrying handbags and backpacks, walking both individually and in groups. The scene also features a few potted plants placed along the sidewalk, adding some greenery to the urban environment. Traffic lights are visible, controlling the flow of vehicles and ensuring the safety of pedestrians.

**Caption with $\varepsilon = 1$:**
The scene unfolds on a bustling city street, teeming with people and vehicles. There are numerous individuals walking along the sidewalk, some closer to the foreground, while others are further away. The street is filled with a variety of vehicles, including cars, [trucks], and taxis. One car is positioned towards the left side of the scene, while another is located more towards the center. Two [trucks] are also present, one near the middle of the street and the other towards the right side. Taxis are a common sight on this busy street, with one taxi near the middle of the scene and another towards the right side of the street. The cityscape is further enhanced by the presence of a fire hydrant on the sidewalk, adding to the urban atmosphere. The scene captures the essence of a typical day in a bustling city, with people and vehicles navigating their way through the streets.

Table 12: Example captions generated with HallE-Switch$_{13B}$.