# OpenReview forum: "HallE-Switch: Rethinking and Controlling Object Existence Hallucinations in Large Vision-Language Models for Detailed Caption"
_ICLR.cc/2024/Conference — Submitted to ICLR 2024_

### Official Review · Reviewer_PjV9 · 2023-10-28

**Soundness:** 3 good
**Presentation:** 3 good
**Contribution:** 2 fair
**Rating:** 3
**Confidence:** 4

**Summary:**

The paper analyzes object hallucination (generating non-existent objects) in detailed image captioning by large vision-language models (LVLMs). It introduces a new evaluation method called CCEval to specifically assess object existence hallucination in detailed captions. Experiments reveal that even LVLMs with minimal hallucination on VQA-based benchmarks show substantial hallucination when evaluated on CCEval.

The paper conducts an analysis attributing hallucination to factors like language decoder size, training data amount/quality, and input image resolution to the vision encoder. The core issue is misalignment between objects mentioned in the caption versus those grounded by the vision encoder. Objects not grounded form incorrect word associations leading to hallucination.

To control hallucination, the paper presents HallE-Switch - an LVLM that can adjust the extent of hallucination via a control parameter. It is trained on datasets with only grounded objects versus with hallucinated objects marked. At inference, the parameter shifts the model between using solely grounded objects (-1) versus blending in hallucinated ones (+1). This achieves 44% hallucination reduction without impacting object coverage or sentence length.

**Strengths:**

1. The writing is clear. I like the flow of this paper where analysis of OH is conducted before providing any solutions.
2. The paper has thorough analysis of factors influencing object hallucination using the new evaluation methods.
2. It is novel to control hallucination levels in LVLMs via contextual/parametric knowledge.
3. The proposed solution maintains object coverage and sentence length while reducing hallucination.

**Weaknesses:**

1. Although it is interesting to argue that not all hallucination is bad, I don't think the authors successfully supported the argument with examples showing when hallucination is beneficial. With that said, more visualizations like example captions may help better explain the hallucination behavior.
2. There could be more specific illustrations on how the training data was generated using GPT-4.
3. Related work section doesn't really provide any useful information connecting existing work and the proposed work. For example, some references are missing such as https://arxiv.org/pdf/2110.01705.pdf.

**Questions:**

I don't have questions in addition to the points mentioned in the weaknesses section.

---

> ### Author Response · Authors · 2023-11-22
> **Reply to Reviewer PjV9 (1/1)**
>
> **Thanks for your feedback! We have largely updated Appendix and Figures to incorporate your suggestions. We also updated the related work section to discuss additional LVLMs. Here, we address the major concerns as following:**
>
>
> **1. Although it is interesting to argue that not all hallucination is bad, I don't think the authors successfully supported the argument with examples showing when hallucination is beneficial. With that said, more visualizations like example captions may help better explain the hallucination behavior.**
>
> We added a section A.1 in Appendix page 1 to support our argument around benefits from parametric knowledge which may also be called “hallucination”. In our study, we define "object existence hallucination" to be a phenomenon where a description makes reference to objects that are not present in the image. However, such hallucinations, when properly harnessed, can be regarded as instances of imagination. Human beings frequently use imagination to successfully accomplish tasks, often without even realizing it.
>
>
> **2. There could be more specific illustrations on how the training data was generated using GPT-4.**
>
> From our understanding, the reviewer needs an illustrations for the data generation process in our method. We update Figure 2 in page 6 to show how we use an open vocabulary detection model to filter objects in ground truth, and recreate detailed caption data based on the filtered objects with GPT4. For how to use GPT4 to generate detailed caption data, we followed a method from LLaVA, by using objects, corresponding bounding boxes, and short captions to generate detailed captions.
>
> **3. Related work section doesn't really provide any useful information connecting existing work and the proposed work. For example, some references are missing such as [1].**
>
> We rewrite the related work and notably cite the paper you mentioned. We find this paper proposes the that label distribution may influence hallucination in image caption, which is much earlier than POPE [2] and LRV [3]. It reduces caption hallucination by balancing ground truth object labels, and adding detection labels into language models, which is really impressive to raise the idea before LLMs were widely used. Our work looks into another aspect which is the alignment issue - hallucination happened in detail captions may be caused by the perceivable-vision and language misalignment. Our finding does not contradict with this previous work that balances the labels distribution to the hallucination in captions. As we can treat balancing objects as a reduction of parametric knowledge, which makes the imagination not so clear and confident. Therefore, it is really nice to see, our work’s findings and experiments can echo previous works and provide new ways to understand the reason for hallucination.
>
> [1] Biten, Ali Furkan, Lluís Gómez, and Dimosthenis Karatzas. "Let there be a clock on the beach: Reducing object hallucination in image captioning." Proceedings of the IEEE/CVF Winter Conference on Applications of Computer Vision. 2022.
>
> [2] Li, Yifan, et al. "Evaluating object hallucination in large vision-language models." arXiv preprint arXiv:2305.10355 (2023).
>
> [3] Liu, Fuxiao, et al. "Aligning Large Multi-Modal Model with Robust Instruction Tuning." arXiv preprint arXiv:2306.14565 (2023).

---

### Official Review · Reviewer_DU9c · 2023-10-29

**Soundness:** 2 fair
**Presentation:** 2 fair
**Contribution:** 2 fair
**Rating:** 5
**Confidence:** 5

**Summary:**

This paper analyzed the cause of hallucination in large vision-language models through the direction of the sizes of large language model, data volume, and input image resolution. The paper further proposes a way to control the generation of VLM, by learning a matrix for mode switching.

**Strengths:**

1. The paper proposes a new benchmark for evaluating hallucination in large vision-language models and analyzes that. Some findings in this paper are interesting and might provide insights for future research.
2. The paper proposes a way to control the hallucination in large vision-language model and obtains improvements on the proposed benchmark.

**Weaknesses:**

1. The overall story is not very coherent. First, the details of CCEval are not very clearly described. Then, analysis is conducted on two or three methods with some conclusions drawn. However, the observation mentioned in the paper seems not to have a specific relation with the proposed Hallu-Switch method. The technique is also only evaluated on CCEval, but previous benchmarks are discussed and used in this paper. The reviewer would expect more insights or explanations about why Hallu-Switch works.
2. The study mainly focuses on LLaVA and InstructBLIP and draws some conclusions for large vision-language models. It might be better to study more models to verify the findings.
3. There are many typos in sentences that hinders the reading and understanding. The paper needs careful revision to fix these issues.
    1. 'We additionally record **and and** balance the average number of objects and the average length of captions across all cases' in the last third paragraph of page 4
    2. ' We find **infeasible** to comparing object hallucinations is impractical when there is a significant disparity in average sentence length and the number of objects.' in the last fourth paragraph of page 4
    3. Table 4, the second column for CCEval should be 'finetuning data' rather than 'model'
    4. 'The learned M can be regarded as the transformation from a generic word space to the object sensitive word space' in the first paragraph of Sec. 3.2. It seems this describes $W$ rather than $M$
4. Small issue that does not affect the rating. Some LVLMs can also be discussed:
    1. Chatspot: Bootstrapping multimodal llms via precise referring instruction tuning
    2. GPT4RoI: Instruction Tuning Large Language Model on Region-of-Interest
    3. MultiModal-GPT: A Vision and Language Model for Dialogue with Humans

**Questions:**

1. The paper mainly discusses LLaVA and InstructBLIP; what if more models are analyzed? Do these findings still holds somehow?

---

> ### Author Response · Authors · 2023-11-22
> **Reply to Reviewer DU9c (1/2)**
>
> **Thanks for your feedback! We have fixed the typos and double checked the paper again. We also updated the related work section to discuss your suggestion on additional LVLMs. Here, we address the major concerns as following:**
>
> **1. The overall story is not very coherent. First, the details of CCEval are not very clearly described. Then, analysis is conducted on two or three methods with some conclusions drawn. However, the observation mentioned in the paper seems not to have a specific relation with the proposed Hallu-Switch method. The technique is also only evaluated on CCEval, but previous benchmarks are discussed and used in this paper.**
>
> We frame our overall story as defining, analyzing, and addressing the problem.
> 1. _How defining:_ Benchmarks that are based on VQA cannot reflect object hallucination for detailed caption. Models did a great job in MME, POPE, does not promise detailed captions without hallucination. We propose a new evaluation metric, CCEval, to measure object existence hallucination reliability.
>
> 2. _How defining leads to our analysis:_ Using CCEval, we can evaluate LVLMs by controlling components to find out the cause of hallucination; _How analysis:_ We analysis language decoder, resolution, and data size to conclude that misalignment between perceivable object in image and object caption form parametric knowledge, which is hallucination.
>
> 3. _How analysis leads to our approach:_ We can decrease the amount of parametric knowledge within the model to reduce hallucination; _How our approach addresses the problem:_ We use HallE-Switch to control the contextual and parametric knowledge.
>
>
> **2. Why technique is only evaluated on CCEval, but previous benchmarks are discussed and used in this paper?**
>
> Technique is only evaluated on CCEval because benchmarks based on VQA cannot reflect object hallucination for detailed caption. Models did a great job in MME, POPE (benchmark for evaluating object existence hallucination by using VQA task), does not promise detailed captions without hallucination. Previous benchmarks are compared with CCEval for showing that model performance on VQA-based benchmark is not correlated to performance on CCEval in the analysis part, so that we can only adhere to CCEval when it comes to our technique, which controls hallucination specifically for detailed captions.
>
>
> **3. The reviewer would expect more insights or explanations about why HallE-Switch works.**
>
> We provide theoretical explanations on why HallE-Switch works in A.9 in Appendix page 5 and 6. In the new version, we updated the section for readability. Here is a short summarization of why it works:
>
> Assuming LLM is good enough to represent an equivalent distribution with Hidden Markov Chain, then there exists an matrix $W$, so that after transferring word embedding $E$ to $WE$, the LLM's originally simulate the text distribution starting with the initial state $\pi$ will turn out to be equivalent to a different initial state. Corresponding to our analysis, the switch method changes the model from a language model with parametric knowledge to a model cutting the parametric knowledge.

---

> > ### Author Response · Authors · 2023-11-22
> > **Reply to Reviewer DU9c (2/2)**
> >
> > **4. The study mainly focuses on LLaVA and InstructBLIP and draws some conclusions for large vision-language models. It might be better to study more models to verify the findings.**
> >
> > We mainly focus on LLaVA and InstructBLIP because they provide multiple language decoder and vision encoder checkpoints, as well as the training data, so that we can break down each component easily to control variables. To evaluate other LVLMs, it would require us to train their models with different language decoder, vision encoders with different image resolution. One challenge is that it is not possible to do so without them releasing the data. Another challenge is that we would incur unfairness if we re-train the models.
> >
> > We try to find some models which release intermediate checkpoints and have different architecture from LLaVA for comparison. So far, we only find MiniGPT4_v1 provides instruction data.
> >
> > | Models        | CHAIR_s           | CHAIR_i | Coverage | Avg. Length | Avg. Object |
> > | ------------- |:-------------:| -----:| -----:| -----:| -----:|
> > | MiniGPT4-7B     | 91.00 | 32.82 | 36.43 | 98.31 | 11.70 | 11.7 |
> > | MiniGPT4-13B      | 89.00     |  34.28 | 34.66 | 125.25 | 12.63 |
> > | MiniGPT4-7B w/ ind. | 85.00    |   27.99 | 32.03 | 151.94 | 14.11 |
> >
> > From the table, we can see MiniGPT4-7B has better performance compared with 13B. This is aligned with our finding that increasing language decoder does not always lead to performance improvement. The reason of this may be that MiniGPT-4 only tunes Q-former using small amount instruction data. Then, we test our finding on the vision and data side - whether aligning objects grounded by vision encoder and objects in training captions can help reduce hallucination? In MiniGPT4-7B w/ ind., we put undetected objects in "[ ]" for training data. The hallucination is reduced and coverage is maintained, which supports our finding.

---

### Official Review · Reviewer_ZWbF · 2023-11-03

**Soundness:** 3 good
**Presentation:** 3 good
**Contribution:** 3 good
**Rating:** 8
**Confidence:** 4

**Summary:**

To tackle the hallucination, the authors introduce CCEval, a novel evaluation method assisted by GPT-4, specifically designed for assessing detailed captioning. Surprisingly, the study reveals that LVLMs exhibit minimal object existence hallucinations in existing Visual Question Answering (VQA) benchmarks. However, the proposed evaluation method exposes continued susceptibility to such hallucinations.

The paper delves into the investigation of these hallucinations and attributes them to various factors, including image resolution, the size of the language decoder, and the quantity, quality, and granularity of instruction data. One of the key findings highlights that hallucinations often occur when the language description includes finer object granularity than what the vision module can ground or verify, leading to unwarranted inferences.

To mitigate these hallucinations, the authors introduce HallE-Switch, a controllable LVLM that addresses object existence hallucinations. This novel approach allows captioning to shift between two modes: (i) exclusively depicting contextual knowledge for grounded objects and (ii) blending contextual knowledge with parametric knowledge to imagine inferred objects. HallE-Switch significantly reduces hallucinations, with a 44% reduction compared to the previous model LLaVA7B, while maintaining the same level of object coverage.

In summary, the paper introduces a new evaluation method, identifies factors contributing to object existence hallucinations in LVLMs, and presents HallE-Switch, a solution that effectively reduces hallucinations in detailed captioning without compromising object coverage. This research contributes to improving the reliability and accuracy of large vision-language models in fine-grained visual description tasks.

**Strengths:**

1.The paper is well-motivated and well designed
2. The proposed method is easy to follow

**Weaknesses:**

1. Some related methods have not been reviewed, such as ``Evaluation and Analysis of Hallucination in Large Vision-Language Models''

**Questions:**

na

---

> ### Author Response · Authors · 2023-11-22
> **Reply to Reviewer ZWbF (1/1)**
>
> Thanks for your feedback! We updated the related work section to reflect your suggestions. HaELM is a LLM-based hallucination evaluation framework, which uses LLM trained on human annotations, and can evaluate a hallucination in image captions, with low cost. This work can make our CCEval  GPT-4 free. I think this work considers realistic things for academic researchers. In the future, we will consider to follow this work to decrease the cost of CCEval.

---

### Official Review · Reviewer_6XKb · 2023-11-04

**Soundness:** 3 good
**Presentation:** 3 good
**Contribution:** 4 excellent
**Rating:** 6
**Confidence:** 4

**Summary:**

The paper studies the problem of object hallucination in large vision-language models detailed captioning. First, the authors quantify the degree of object hallucination by varying model size, fine-tuning data size, image resolution, etc. Second, they proposed two methods, namely (1) modifying the caption training data to distinguish contextual object vs parametric object; (2) adding a layer that acts as a switch between contextual knowledge and parametric knowledge. The proposed methods outperforms baseline on CCEval benchmark.

**Strengths:**

- Addressed a very timely topic of object hallucination in large vision-language models.
- Interpreting the objects using contextual/parametric knowledge framework seems novel.
- Showed results on multiple architectures, model scales, amount of fine-tune data, which are valuable findings to the community's future research.

**Weaknesses:**

- Presentation needs polishing. The paper is very dense in text and requires some effort for reading.  Also, please address the points in "Questions" section.
- The first part and second part of the paper looks more like two separate and condensed papers. Due to this problem, especially the first part fails to deepen our insight about the root cause of the problem. I would expect a deeper treatment on each part.
- Results are mostly quantitative. It would be better to show more qualitative examples of hallucination.

**Questions:**

I'm generally happy with this submission and think it is above the acceptance bar. Readers could appreciate this work better if the presentation is improved. Please answer the following minor points.

1. In page 4, how are "consistent constraints" enforced? Please explain in detail.
2. Section 3.2 is not very clear to me. Does W correspond to "Projector" in Fig 2? According to Fig 2, W comes after LLM. However, equation   says the opposite. Is the epsilon parameter applied on word-by-word basis or sentence-by-sentence basis?
I may have misunderstood something because I'm not familiar with the LM-Switch work. Regardless, I believe a good paper should be self-contained and can be readable to general audience in the community.
3. In page 2, object relationship hallucination is mentioned but this concept does not seem to appear again later pages, in metrics or methods presented in the paper. Did I misunderstood?
4. Do you observe any downsides or limitations of using this method that were not expressed in the result Table?

---

> ### Author Response · Authors · 2023-11-22
> **Reply to Reviewer 6XKb (1/2)**
>
> **Thanks for your feedback! We have updated the paper for better presentation. Here, we address the major concerns as following:**
>
> **- Presentation needs polishing. The paper is very dense in text and requires some effort for reading. Also, please address the points in "Questions" section.**
>
> Please refer to the Questions section.
>
> **- The first part and second part of the paper looks more like two separate and condensed papers. Due to this problem, especially the first part fails to deepen our insight about the root cause of the problem. I would expect a deeper treatment on each part.**
>
> We refer the reviewer to revised Section 2 Conclusion in page 6 where it summarizes the entire first part and proposes an explanation on the root cause of the problem. We also revised Section 3 Intro in page 6 to better connect with Section 2. In short, the root cause is the misalignment between perceivable vision and language data, developing contextual and parametric knowledge, which is hallucination. We also refer the reviewer to our first reply for reviewer DU9c for a run through of the overall story.
>
> **- Results are mostly quantitative. It would be better to show more qualitative examples of hallucination.**
>
> We add a section A.8 in Appendix page 5 to show the caption differences with different switch values on 5 images. The qualitative results align with the quantitative ones.
> ## Questions:
> **1. In page 4, how are "consistent constraints" enforced?**
>
> We want to clarify that we do not strictly enforce consistent constraints, neither this is realistic to achieve. Although our intention is to make the model comparable and push the model to express more objects, it is sometimes not possible for some of the models, such as BLIP2, to express detailed captions, and our coverage, average length, and average number of object metrics in CCEval can accurately reflect if any model has such limitations.
>
> In the paper, "consistent constraints'' are mainly achieved through prompting. We use the same prompt - “describe this image in detail” - across all four models in Table 2. Simply prompting can result in similar length and number of objects captions in this case since the instruction finetuning data for detailed captioning all comes from a similar source (LLaVA synthetic data). For models with different instruction finetuning data, we suggest further adjustment of the prompt or set the min_length and max_length in the inference function.
>
> **2. Section 3.2 is not very clear to me. Does W correspond to "Projector" in Fig 2? According to Fig 2, W comes after LLM. However, equation says the opposite. Is the epsilon parameter applied on word-by-word basis or sentence-by-sentence basis?**
>
> To explain, $W$ is a projector and it is inserted between LLM backbone and LLM head. Therefore, we have to pass the projector into $M$, where $M$ is LLM backbone + head, resulted in the equation $M’ = M(eps*W)$. We update the Figure 2 and Section 3.2 in page 6 and 7. In the new version, we separately represent LLM backbone as $B$ and LLM head as $H$ in both equation and figure, so it should be clearer.
>
> The epsilon parameter is the same value for every word in one caption and multiplied with the projector W. Therefore, the answer is word-by-word basis since the model processes each word, but it is the same for every word in one caption.
>
> **3. In page 2, object relationship hallucination is mentioned but this concept does not seem to appear again later pages, in metrics or methods presented in the paper. Did I misunderstood?**
>
> In this work, we intentionally only focus on object existence hallucination as mentioned in Introduction. We consider object attributes and relationship hallucination as our future works. The reason is that - it is not surprising that existing LVLMs fail on expressing object attribute and relationship, because MSCOCO dataset does not include this information, and existing LVLMs’ caption data heavily rely on this dataset. However, it is surprising that object existence hallucination is still an issue for these models, because current detailed caption training data is accurate in object existence. Therefore, we decided to tackle this problem first.

---

> ### Author Response · Authors · 2023-11-22
> **Reply to Reviewer 6XKb (2/2)**
>
> **4. Do you observe any downsides or limitations of using this method that were not expressed in the result Table?**
>
> One of the limitations is that the method reaches the best performance with larger image resolution, which aligns with our finding. However, increasing image resolution inevitably leads to computation cost.
>
> Another limitation is the extrapolation ability of the switch. Since we train with epsilon +1 and -1, it makes sense to interpolate well when epsilon is between +1 and -1 during inference, and our Table 8 result on CCEval meet this expectation. However, we thought it would be very interesting to test the extrapolation ability of the switch. Specifically, given a switch is trained using epsilon +1 and -1, can we use epsilon < -1 during inference? It would be very nice if the hallucination continued to decrease for epsilon < -1. Unfortunately, hallucination did not decrease further with -1.2, -1.5. In addition, the model starts to repeat and loss language ability if epsilon is outside of range [1, -1]:
>
> _Example captions with switch value -1.2:_ The scene unfolds on a dining table, which is neatly arranged with a variety of food items. There are two plates of food, one containing a sandwich and the other with a piece of cake. The sandwich is located towards the left side of the table, while the cake is positioned towards the right. In addition to the food, there are several utensils scattered across the table. A knife is located towards the left side of the table, while a fork is positioned towards the right. There are also two spoons, one near the center of the table and the other towards the right side. (The table is not just for food; there are also several books scattered across the surface. One book is located towards the left side of the table, while another is positioned towards the right. The third book is located towards the center of the table.) x N times
>
> _Example captions with switch value -1.5:_ The scene unfolds on a bustling city street, where a man is walking across the street. He is wearing a backpack and appears to be in a hurry. There are several other people scattered around the scene, each engaged in their own activities. Starting from the left side of the scene, there's a car parked on the side of the street. Moving towards the right, there's another car, followed by a truck. Further to the right, there's a traffic light, indicating the direction of the flow of traffic. Starting from the top of the scene, there's a person standing near the car on the left side. Moving down, there's another person standing near the car on the right side. Further down, there's a person standing near the truck. (Starting from the top of the scene, there's another person standing near the car on the left side. Moving down, there's another person standing near the car on the right side.) x N times

---

> > ### Comment · Reviewer_6XKb · 2023-11-23
> >
> > I appreciate the authors for preparing a lengthy rebuttal. I have also reviewed the responses to other reviewers' questions. I still think the work should be accepted.

---

### Author Response · Authors · 2023-11-22
**Reply to All Reviewers**

We sincerely appreciate the reviewers’ comprehensive suggestions. We are glad to see the reviewers find our topic well-motivated and timely (6XKb, ZWbF), method and framework novel and performant (6XKb, DU9c, PjV9), writing clear (ZWbF, PjV9), and provide insightful findings to the community (6XKb, DU9c).

In this work, we first provide an evaluation method for object existence hallucination in detailed captions. Using this metric, we analyze each component of LVLM to find the cause of hallucination. The cause is the misalignment between perceivable vision and language data develops contextual and parametric knowledge (hallucinations). Finally, we propose a method to effectively control contextual and parametric knowledge.

In the updated version, blue text indicates major revision. Specifically, we address the minor confusion on our paper scope in Introduction, add more details on CCEval in Section 2.1, revise the analysis conclusion and method first part to better transition from hallucination analysis to controlling in Section 3. Further, we update method formulation due to minor confusion in Section 3.2. Most importantly, we update the figures for better presentation and rewrite the related work section to provide more insights. We also add multiple sections in Appendix. We strongly recommend reviewers to refer to our Appendix because it contains useful visualizations that cannot fit in the main paper.

---

### Meta-Review · Area_Chair_8V3S · 2024-01-05

**Metareview:**

The authors focus on the phenomenon of object hallucination of current large vision-language models when applied to image captioning tasks. They note that current benchmarks may under-diagnose the problem of object hallucination in generated captions, and propose a new, GPT-4 assisted, evaluation method (CCEval) that is tailored to the assessment of evaluations. Using CCEval, they investigate to what degree aspects like image resolution, language decoder size, etc. impact hallucinations. Finally, they introduce a method called HallE-Switch, designed to allow control of hallucinations.

Reviews highlighted strengths, including that the paper addresses a very timely topic from a novel perspective, that the paper is well-motivated, and that it examines multiple aspects that may impact hallucinations. They also note the positive empirical results. At the same time, multiple weaknesses were highlighted, including: concerns on whether the authors' argument that not all hallucination is bad was sufficiently supported with the provided illustrative examples, questions on more specifically illustrating how GPT-4 was used in the evaluation, clarity of describing and validating CCEval, thoughts on whether more models should be included in such a study, lack of some relevant papers in the related work section, and general clarity and presentation issues.

The authors followed up with a detailed rebuttal, including additional empirical analysis and revision of the manuscript. The rebuttal was discussed in author-reviewer as well as reviewer-AC discussions.

At the end of the rebuttal period, reviewers are split, with two reviewers in support and two with remaining reservations. After follow up discussion with reviewers, the AC agrees with the reviewers who still hold reservations. In particular, the AC agrees with reviewers noting that the proposed evaluation method CCEval is not validated in sufficient depth. The recommendation is to reject the manuscript in its current form.

**Justification For Why Not Higher Score:**

The paper introduces a form of evaluation that is not sufficiently documented and validated.

**Justification For Why Not Lower Score:**

N/A

---

### Decision · Program_Chairs · 2024-01-16

Reject